# Recent quantitative research on determinants of health in high income countries: A scoping review

**Vladimira Varbanova** *, **Philippe Beutels**

Centre for Health Economics Research and Modelling Infectious Diseases, Vaccine and Infectious Disease Institute, University of Antwerp, Antwerp, Belgium

* vladimira.varbanova@uantwerpen.be

## Abstract

### Background

Identifying determinants of health and understanding their role in health production constitutes an important research theme. We aimed to document the state of recent multi-country research on this theme in the literature.

### Methods

We followed the PRISMA-ScR guidelines to systematically identify, triage and review literature (January 2013—July 2019). We searched for studies that performed cross-national statistical analyses aiming to evaluate the impact of one or more aggregate level determinants on one or more general population health outcomes in high-income countries. To assess in which combinations and to what extent individual (or thematically linked) determinants had been studied together, we performed multidimensional scaling and cluster analysis.

### Results

Sixty studies were selected, out of an original yield of 3686. Life-expectancy and overall mortality were the most widely used population health indicators, while determinants came from the areas of healthcare, culture, politics, socio-economics, environment, labor, fertility, demographics, life-style, and psychology. The family of regression models was the predominant statistical approach. Results from our multidimensional scaling showed that a relatively tight core of determinants have received much attention, as main covariates of interest or controls, whereas the majority of other determinants were studied in very limited contexts. We consider findings from these studies regarding the importance of any given health determinant inconclusive at present. Across a multitude of model specifications, different country samples, and varying time periods, effects fluctuated between statistically significant and not significant, and between beneficial and detrimental to health.

### Conclusions

We conclude that efforts to understand the underlying mechanisms of population health are far from settled, and the present state of research on the topic leaves much to be desired. It

**Data Availability Statement:** All relevant data are within the manuscript and its Supporting Information files.

**Funding:** This study (and VV) is funded by the Research Foundation Flanders (https://www.fwo.

be/en/), FWO project number G0D5917N, award obtained by PB. The funder had no role in study design, data collection and analysis, decision to publish, or preparation of the manuscript.

**Competing interests:** The authors have declared that no competing interests exist.

is essential that future research considers multiple factors simultaneously and takes advantage of more sophisticated methodology with regards to quantifying health as well as analyzing determinants' influence.

## Introduction

Identifying the key drivers of population health is a core subject in public health and health economics research. Between-country comparative research on the topic is challenging. In order to be relevant for policy, it requires disentangling different interrelated drivers of "good health", each having different degrees of importance in different contexts.

"Good health"–physical and psychological, subjective and objective–can be defined and measured using a variety of approaches, depending on which aspect of health is the focus. A major distinction can be made between health measurements at the individual level or some aggregate level, such as a neighborhood, a region or a country. In view of this, a great diversity of specific research topics exists on the drivers of what constitutes individual or aggregate "good health", including those focusing on health inequalities, the gender gap in longevity, and regional mortality and longevity differences.

The current scoping review focuses on determinants of population health. Stated as such, this topic is quite broad. Indeed, we are interested in the very general question of what methods have been used to make the most of increasingly available region or country-specific databases to understand the drivers of population health through inter-country comparisons. Existing reviews indicate that researchers thus far tend to adopt a narrower focus. Usually, attention is given to only one health outcome at a time, with further geographical and/or population [1, 2] restrictions. In some cases, the impact of one or more interventions is at the core of the review [3–7], while in others it is the relationship between health and just one particular predictor, e.g., income inequality, access to healthcare, government mechanisms [8–13]. Some relatively recent reviews on the subject of social determinants of health [4–6, 14–17] have considered a number of indicators potentially influencing health as opposed to a single one. One review defines "social determinants" as "the social, economic, and political conditions that influence the health of individuals and populations" [17] while another refers even more broadly to "the factors apart from medical care" [15].

In the present work, we aimed to be more inclusive, setting no limitations on the nature of possible health correlates, as well as making use of a multitude of commonly accepted measures of general population health. The goal of this scoping review was to document the state of the art in the recent published literature on determinants of population health, with a particular focus on the *types of determinants* selected and the *methodology* used. In doing so, we also report the main characteristics of the results these studies found. The materials collected in this review are intended to inform our (and potentially other researchers') future analyses on this topic. Since the production of health is subject to the law of diminishing marginal returns, we focused our review on those studies that included countries where a high standard of wealth has been achieved for some time, i.e., high-income countries belonging to the Organisation for Economic Co-operation and Development (OECD) or Europe. Adding similar reviews for other country income groups is of limited interest to the research we plan to do in this area.

## Methods

In view of its focus on data and methods, rather than results, a formal protocol was not registered prior to undertaking this review, but the procedure followed the guidelines of the PRISMA statement for scoping reviews [18].

## Search

We focused on multi-country studies investigating the potential associations between any aggregate level (region/city/country) determinant and general measures of population health (e.g., life expectancy, mortality rate).

Within the query itself, we listed well-established population health indicators as well as the six world regions, as defined by the World Health Organization (WHO). We searched only in the publications' titles in order to keep the number of hits manageable, and the ratio of broadly relevant abstracts over all abstracts in the order of magnitude of 10% (based on a series of time-focused trial runs). The search strategy was developed iteratively between the two authors and is presented in S1 Appendix. The search was performed by VV in PubMed and Web of Science on the 16th of July, 2019, without any language restrictions, and with a start date set to the 1st of January, 2013, as we were interested in the latest developments in this area of research.

## Eligibility criteria

Records obtained via the search methods described above were screened independently by the two authors. Consistency between inclusion/exclusion decisions was approximately 90% and the 43 instances where uncertainty existed were judged through discussion. Articles were included subject to meeting the following requirements: (a) the paper was a full published report of an original empirical study investigating the impact of at least one aggregate level (city/region/country) factor on at least one health indicator (or self-reported health) of the general population (the only admissible "sub-populations" were those based on gender and/or age); (b) the study employed statistical techniques (calculating correlations, at the very least) and was not purely descriptive or theoretical in nature; (c) the analysis involved at least two countries or at least two regions or cities (or another aggregate level) in at least two different countries; (d) the health outcome was not differentiated according to some socio-economic factor and thus studied in terms of inequality (with the exception of gender and age differentiations); (e) mortality, in case it was one of the health indicators under investigation, was strictly "total" or "all-cause" (no cause-specific or determinant-attributable mortality).

## Data extraction

The following pieces of information were extracted in an Excel table from the full text of each eligible study (primarily by VV, consulting with PB in case of doubt): health outcome(s), determinants, statistical methodology, level of analysis, results, type of data, data sources, time period, countries. The evidence is synthesized according to these extracted data (often directly reflected in the section headings), using a narrative form accompanied by a "summary-of-findings" table and a graph.

## Results

### Search and selection

The initial yield contained 4583 records, reduced to 3686 after removal of duplicates (Fig 1). Based on title and abstract screening, 3271 records were excluded because they focused on specific medical condition(s) or specific populations (based on morbidity or some other factor), dealt with intervention effectiveness, with theoretical or non-health related issues, or with animals or plants. Of the remaining 415 papers, roughly half were disqualified upon full-text consideration, mostly due to using an outcome not of interest to us (e.g., health inequality), measuring and analyzing determinants and outcomes exclusively at the individual level,

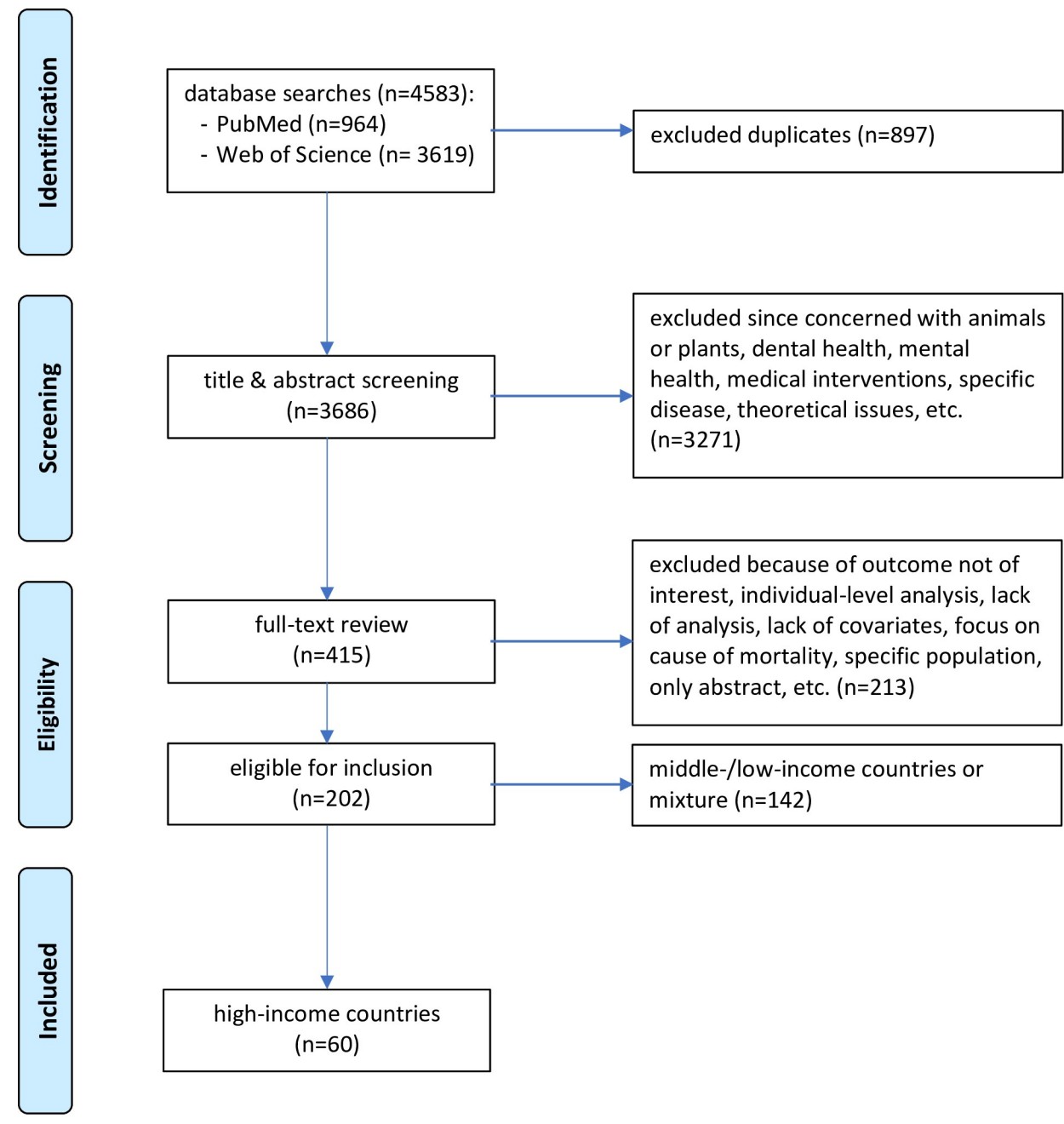

**Fig 1. PRISMA flow-diagram.**

performing analyses one country at a time, employing indices that are a mixture of both health indicators and health determinants, or not utilizing potential health determinants at all. After this second stage of the screening process, 202 papers were deemed eligible for inclusion. This group was further dichotomized according to level of economic development of the countries or regions under study, using membership of the OECD or Europe as a reference "cut-off" point. Sixty papers were judged to include high-income countries, and the remaining 142 included either low- or middle-income countries or a mix of both these levels of development. The rest of this report outlines findings in relation to high-income countries only, reflecting

our own primary research interests. Nonetheless, we chose to report our search yield for the other income groups for two reasons. First, to gauge the relative interest in applied published research for these different income levels; and second, to enable other researchers with a focus on determinants of health in other countries to use the extraction we made here.

## Health outcomes

The most frequent population health indicator, life expectancy (LE), was present in 24 of the 60 studies. Apart from "life expectancy at birth" (representing the average life-span a newborn is expected to have if current mortality rates remain constant), also called "period LE" by some [19, 20], we encountered as well LE at 40 years of age [21], at 60 [22], and at 65 [21, 23, 24]. In two papers, the age-specificity of life expectancy (be it at birth or another age) was not stated [25, 26].

Some studies considered male and female LE separately [21, 24, 25, 27–33]. This consideration was also often observed with the second most commonly used health index [28–30, 34–38]–termed "total", or "overall", or "all-cause", mortality rate (MR)–included in 22 of the 60 studies. In addition to gender, this index was also sometimes broken down according to age group [30, 39, 40], as well as gender-age group [38].

While the majority of studies under review here focused on a single health indicator, 23 out of the 60 studies made use of multiple outcomes, although these outcomes were always considered one at a time, and sometimes not all of them fell within the scope of our review. An easily discernable group of indices that typically went together [25, 37, 41] was that of neonatal (deaths occurring within 28 days postpartum), perinatal (fetal or early neonatal / first-7-days deaths), and post-neonatal (deaths between the 29th day and completion of one year of life) mortality. More often than not, these indices were also accompanied by "stand-alone" indicators, such as infant mortality (deaths within the first year of life; our third most common index found in 16 of the 60 studies), maternal mortality (deaths during pregnancy or within 42 days of termination of pregnancy), and child mortality rates. Child mortality has conventionally been defined as mortality within the first 5 years of life, thus often also called "under-5 mortality". Nonetheless, Pritchard & Wallace used the term "child mortality" to denote deaths of children younger than 14 years [42].

As previously stated, inclusion criteria did allow for self-reported health status to be used as a general measure of population health. Within our final selection of studies, seven utilized some form of subjective health as an outcome variable [25, 43–48]. Additionally, the Health Human Development Index [49], healthy life expectancy [50], old-age survival [51], potential years of life lost [52], and disability-adjusted life expectancy [25] were also used.

We note that while in most cases the indicators mentioned above (and/or the covariates considered, see below) were taken in their absolute or logarithmic form, as a—typically annual—number, sometimes they were used in the form of differences, change rates, averages over a given time period, or even z-scores of rankings [19, 22, 40, 42, 44, 53–57].

## Regions, countries, and populations

Despite our decision to confine this review to high-income countries, some variation in the countries and regions studied was still present. Selection seemed to be most often conditioned on the European Union, or the European continent more generally, and the Organisation of Economic Co-operation and Development (OECD), though, typically, not all member nations–based on the instances where these were also explicitly listed—were included in a given study. Some of the stated reasons for omitting certain nations included data unavailability [30, 45, 54] or inconsistency [20, 58], Gross Domestic Product (GDP) too low [40], differences in economic development and political stability with the rest of the sampled countries

[59], and national population too small [24, 40]. On the other hand, the rationales for selecting a group of countries included having similar above-average infant mortality [60], similar healthcare systems [23], and being randomly drawn from a social spending category [61]. Some researchers were interested explicitly in a specific geographical region, such as Eastern Europe [50], Central and Eastern Europe [48, 60], the Visegrad (V4) group [62], or the Asia/ Pacific area [32]. In certain instances, national regions or cities, rather than countries, constituted the units of investigation instead [31, 51, 56, 62–66]. In two particular cases, a mix of countries and cities was used [35, 57]. In another two [28, 29], due to the long time periods under study, some of the included countries no longer exist. Finally, besides "European" and "OECD", the terms "developed", "Western", and "industrialized" were also used to describe the group of selected nations [30, 42, 52, 53, 67].

As stated above, it was the health status of the *general* population that we were interested in, and during screening we made a concerted effort to exclude research using data based on a more narrowly defined group of individuals. All studies included in this review adhere to this general rule, albeit with two caveats. First, as cities (even neighborhoods) were the unit of analysis in three of the studies that made the selection [56, 64, 65], the populations under investigation there can be more accurately described as *general urban*, instead of just general. Second, oftentimes health indicators were stratified based on gender and/or age, therefore we also admitted one study that, due to its specific research question, focused on men and women of early retirement age [35] and another that considered adult males only [68].

## Data types and sources

A great diversity of sources was utilized for data collection purposes. The accessible reference databases of the OECD (https://www.oecd.org/), WHO (https://www.who.int/), World Bank (https://www.worldbank.org/), United Nations (https://www.un.org/en/), and Eurostat (https://ec.europa.eu/eurostat) were among the top choices. The other international databases included Human Mortality [30, 39, 50], Transparency International [40, 48, 50], Quality of Government [28, 69], World Income Inequality [30], International Labor Organization [41], International Monetary Fund [70]. A number of national databases were referred to as well, for example the US Bureau of Statistics [42, 53], Korean Statistical Information Services [67], Statistics Canada [67], Australian Bureau of Statistics [67], and Health New Zealand Tobacco control and Health New Zealand Food and Nutrition [19]. Well-known surveys, such as the World Values Survey [25, 55], the European Social Survey [25, 39, 44], the Eurobarometer [46, 56], the European Value Survey [25], and the European Statistics of Income and Living Condition Survey [43, 47, 70] were used as data sources, too. Finally, in some cases [25, 28, 29, 35, 36, 41, 69], built-for-purpose datasets from previous studies were re-used.

In most of the studies, the level of the data (and analysis) was national. The exceptions were six papers that dealt with Nomenclature of Territorial Units of Statistics (NUTS2) regions [31, 62, 63, 66], otherwise defined areas [51] or cities [56], and seven others that were multilevel designs and utilized both country- and region-level data [57], individual- and city- or country-level [35], individual- and country-level [44, 45, 48], individual- and neighborhood-level [64], and city-region- (NUTS3) and country-level data [65]. Parallel to that, the data type was predominantly longitudinal, with only a few studies using purely cross-sectional data [25, 33, 43, 45–48, 50, 62, 67, 68, 71, 72], albeit in four of those [43, 48, 68, 72] two separate points in time were taken (thus resulting in a kind of "double cross-section"), while in another the averages across survey waves were used [56].

In studies using longitudinal data, the length of the covered time periods varied greatly. Although this was almost always less than 40 years, in one study it covered the entire 20[th]

century [29]. Longitudinal data, typically in the form of annual records, was sometimes transformed before usage. For example, some researchers considered data points at 5- [34, 36, 49] or 10-year [27, 29, 35] intervals instead of the traditional 1, or took averages over 3-year periods [42, 53, 73]. In one study concerned with the effect of the Great Recession all data were in a "recession minus expansion change in trends"-form [57]. Furthermore, there were a few instances where two different time periods were compared to each other [42, 53] or when data was divided into 2 to 4 (possibly overlapping) periods which were then analyzed separately [24, 26, 28, 29, 31, 65]. Lastly, owing to data availability issues, discrepancies between the time points or periods of data on the different variables were occasionally observed [22, 35, 42, 53–55, 63].

## Health determinants

Together with other essential details, Table 1 lists the health correlates considered in the selected studies. Several general categories for these correlates can be discerned, including health care, political stability, socio-economics, demographics, psychology, environment, fertility, life-style, culture, labor. All of these, directly or implicitly, have been recognized as holding importance for population health by existing theoretical models of (social) determinants of health [74–77].

It is worth noting that in a few studies there was just a single aggregate-level covariate investigated in relation to a health outcome of interest to us. In one instance, this was life satisfaction [44], in another–welfare system typology [45], but also gender inequality [33], austerity level [70, 78], and deprivation [51]. Most often though, attention went exclusively to GDP [27, 29, 46, 57, 65, 71]. It was often the case that research had a more particular focus. Among others, minimum wages [79], hospital payment schemes [23], cigarette prices [63], social expenditure [20], residents' dissatisfaction [56], income inequality [30, 69], and work leave [41, 58] took center stage. Whenever variables outside of these specific areas were also included, they were usually identified as confounders or controls, moderators or mediators.

We visualized the combinations in which the different determinants have been studied in Fig 2, which was obtained via multidimensional scaling and a subsequent cluster analysis (details outlined in S2 Appendix). It depicts the spatial positioning of each determinant relative to all others, based on the number of times the effects of each pair of determinants have been studied simultaneously. When interpreting Fig 2, one should keep in mind that determinants marked with an asterisk represent, in fact, collectives of variables.

Distances between determinants in Fig 2 are indicative of determinants' "connectedness" with each other. While the statistical procedure called for higher dimensionality of the model, for demonstration purposes we show here a two-dimensional solution. This simplification unfortunately comes with a caveat. To use the factor smoking as an example, it would appear it stands at a much greater distance from GDP than it does from alcohol. In reality however, smoking was considered together with alcohol consumption [21, 25, 26, 52, 68] in just as many studies as it was with GDP [21, 25, 26, 52, 59], five. To aid with respect to this apparent shortcoming, we have emphasized the strongest pairwise links. Solid lines connect GDP with health expenditure (HE), unemployment rate (UR), and education (EDU), indicating that the effect of GDP on health, taking into account the effects of the other three determinants as well, was evaluated in between 12 to 16 studies of the 60 included in this review. Tracing the dashed lines, we can also tell that GDP appeared jointly with income inequality, and HE together with either EDU or UR, in anywhere between 8 to 10 of our selected studies. Finally, some weaker but still worth-mentioning "connections" between variables are displayed as well via the dotted lines.

**Table 1. List of studies included in the review.**

| | author(s) | region | time span | outcome(s) | covariates* | methods |
|---|---|---|---|---|---|---|
| 1. | Bender, Economou, & Theodossiou (2013) | 11 European countries | 1971–2001 | all-cause M; IM | UR; GDP; %population above 25yo with no education; %population above 25yo with post-secondary diploma | fixed effects regression; feasible generalized least squares |
| 2. | Erdogan, Ener, & Arica (2013) | 25 OECD countries | 1970–2007 | IMR | GDP | fixed effects model |
| 3. | Harding, Lenguerrand, Costa, d'Errico, Martikainen, Tarkiainen, Blane, Akinwale, & Bartley (2013) | 3 European regions | 1971–2001 | all-cause M | welfare regime (liberal, conservative, social democratic) | Poisson regression |
| 4. | Mackenbach (2013) | 40 national European units | 1900–2008 | LEaB | GDP | ordinary least squares linear regression |
| 5. | Mackenbach, Hu, & Looman (2013) | Europe | 1960–2008 | LEaB; all-cause M | revised Polity2 democracy level index; current democracy; cumulative years of democracy; GDP; average years of schooling (for above 25yo); transition to independence; armed conflict; Economic Freedom of the World index | fixed effects ordinary least squares regression |
| 6. | Mackenbach & Looman (2013a) | 25 European countries | 1955–1989 | all-cause M | GDP; Polity2 democracy level index | ordinary least squares linear regression |
| 7. | Mackenbach & Looman (2013b) | WHO European region | 1900–2008 | LEaB; all-cause M | GDP | simple linear regression |
| 8. | Minagawa (2013) | 23 Eastern European countries | 2008–2009 | HLE | Corruption Perceptions Index; economic freedom; societal freedom; freedom of the press; #terrorist attacks in a year; prison population rate; GDP; %total HE | generalized least-squares regression |
| 9. | Asandului, Pintilescu, Jemna, & Viorica (2014) | 8 CEE EU countries | 1989–2012 | IMR | GDP; UR; LEaB; abortion rate; vaccination rate (%children younger than 2yo vaccinated for DPT); public HE; average age of females at first birth | correlations; fixed effects model |
| 10. | Barthold, Nandi, Rodriguez, & Heymann (2014) | 27 OECD countries | 1991–2007 | LEaB; LEa40; LEa65 | HE; social expenditure; GDP; % population above 65yo; education expenditure; %population with tertiary/ upper secondary/primary education; smoking; alcohol consumption | ordinary least squares regression |
| 11. | Baumbach & Gulis (2014) | 8 EU countries | 2000–2010 | overall M | GDP; UR; public social spending | correlations |
| 12. | Lopez-Casasnovas & Soley-Bori (2014) | 32 OECD countries | 1980–2010 | HHDI | GDP; UR; Gini coefficient of wealth inequality; social expenditure; HE; existence of a National Health System | random effects regression |
| 13. | Mackenbach (2014) | 42 European countries | 2010 | LE; DALE; self-assessed health; neonatal M; post-neonatal M; maternal M | GDP; %population in urbanized areas; % daily smokers; alcohol consumption; spirits consumption; teenage pregnancy; %older mothers; 3 groups of cultural values: Inglehart scales—self-expression, secular-rational; Hofstede indices—power distance, individualism, uncertainty avoidance, masculinity, long-term orientation, indulgence; Schwarts orientations—affective autonomy, intellectual autonomy, embeddedness, egalitarianism, hierarchy, harmony, mastery; | Pearson correlations; multivariate linear least squares regression |
| 14. | Megyesiova & Lieskovska (2014) | 28 EU member states | 2005; 2012 | self-reported health status | GDP; final consumption expenditure of household per inhabitant; compensation of employees per inhabitant | Spearman´s rank correlation coefficients |

(*Continued*)

**Table 1.** (Continued)

| | author(s) | region | time span | outcome(s) | covariates* | methods |
|---|---|---|---|---|---|---|
| 15. | Torre & Myrskyla (2014) | 21 developed countries | 1975–2006 | LEaB; MR | Gini index of income inequality; GDP | correlations; fixed effects regression |
| 16. | Budhdeo, Watkins, Atun, Williams, Zeltner, & Maruthappu (2015) | 27 EU countries | 1995–2010 | neonatal M; post-neonatal M; 1-5yo M; <5yrs M; adult M | government HE; population size; % population above 65yo; % population under 15yo; GDP; inflation; UR; government debt; urbanization; mean calorie intake; access to water; out-of-pocket expenditures; #hospital beds; #physicians; private HE | fixed effects regression |
| 17. | Gathmann, Jurges, & Reinhold (2015) | 11 European countries | 1903–1976 | MR | compulsory schooling reform | meta-analysis (of reduced form & 2-sample two stage least squares estimates) |
| 18. | Hu, van Lenthe, & Mackenbach (2015) | 43 European countries | 1987–2008 | LEaB; all-cause M; IMR | Gini index; GDP; democracy indicator; average years of schooling; transition to national independence; armed conflict; economic freedom | fixed effects models |
| 19. | Iacob, Volintiru, Cristea, & Turcu (2015) | 30 European countries | 2013 | LEaB | GDP | least squares regression |
| 20. | Karyani, Kazemi, Shaahmadi, Arefi, & Meshkani (2015) | OECD countries | 2010; 2013 | under 5 M | public HE; GNI; physician density; nurses' density; ratio of female to male primary/secondly/tertiary school enrollment | Pearson correlations; regression |
| 21. | Koots-Ausmees & Realo (2015) | 32 European countries | 2002–2012 | subjective well-being | life satisfaction | correlations |
| 22. | Pritchard & Wallace (2015) | 21 Western countries | 1979–2010 | CMR | HE; income inequality | Spearman rank order correlations |
| 23. | Pritchard, Williams, & Wallace (2015) | 21 Western nations | 1979–2010 | CMR | HE; income inequality | Spearman rank order correlations |
| 24. | Safaei (2015) | 31 OECD countries | 2008–2010 | LEaB; IMR; CMR | pro-primary distribution orientation; pro-secondary distribution orientation; GDP | correlations; ordinary least squares regression |
| 25. | Xie, Gaudet, Krewski, Graham, Walker, & Wen (2015) | 31 industrialized countries | 2010 | IMR | Cesarean delivery rate; maternal age; infant sex ratio; multiple pregnancy; GDP; Gini index; preterm birth rate | Pearson correlation coefficients; multiple linear regression |
| 26. | Zare, Gaskin, & Anderson (2015) | 30 OECD countries | 1985–2010 | LE | GDP; %daily smokers; alcohol consumption; daily Kcal intake; schooling years; fertility rate; %females; labor productivity; greenhouse gas; democracy index; governance index; %employees in industry; public social expenditure | random effects model |
| 27. | Bartoll & Mari-Dell'Olmo (2016) | 232 European regions | 2003–2012 | LEaB | UR; regional income; national social protection typology; gender | 1st differences model |
| 28. | Bremberg (2016) | 28 OECD countries | 1990–2012 | IMR | GDP; labor productivity; Gini index; child income poverty; general government revenues; public spending on family benefits in cash, services and tax measures; public HE; attained tertiary education degree (25–64yo's); adult literacy (prose) score; gross domestic expenditure on research & development; trust; %daily female smokers; "history" variable | least squares linear multiple regression |
| 29. | Shim (2016) | 19 OECD countries | 1969–2010 | IMR; perinatal MR; neonatal MR; post-neonatal MR; CMR | job-protected paid leave; other leave; GDP; total HE; %population covered by health insurance; #kidney dialysis patients; total fertility rates; female employment rates; low birth weight; immunization rates for measles by age 1; immunization rates for DPT by age 1; expenditures on family cash allowances; expenditures on maternity & parental leave; expenditures on family services | fixed effects ordinary least squares regression |

(*Continued*)

**Table 1.** (Continued)

| | author(s) | region | time span | outcome(s) | covariates* | methods |
|---|---|---|---|---|---|---|
| 30. | Wubulihasimu, Brouwer, & van Baal (2016) | 20 OECD countries | 1980–2009 | LEaB; LEa65; IM | hospital payment scheme (fixed budget, fee-for-service, patient-based payment); GDP; %population above 65yo | difference-in-difference |
| 31. | Blazquez-Fernandez, Cantarero-Prieto, & Pascual-Saez (2017) | 8 OECD Asia/ Pacific area countries | 1995–2013 | LEaB | GDP; HE; UR; exchange rate | "panel and time-series data techniques" |
| 32. | Bremberg (2017) | 28 OECD countries | 1990–2010 | MR | GDP; Gini index; average social spending; publicly funded health care; attained tertiary education degree (25–64 yo's); corruption index; historical levels of mortality | multiple regression |
| 33. | Filippidis, Laverty, Hone, Been, & Millett (2017) | 276 subnational regions within 23 EU countries | 2004–2014 | IMR | median cigarette prices; cigarette price differentials; % of 25-64yo population with tertiary education; GDP; UR; % of all births by high risk mothers (age <18 or ≥40yrs); Smoke-free Work and Other Public Places subscale of the Tobacco Control Scale | linear fixed effects regression |
| 34. | Granados & Ionides (2017) | 27 European countries | 1995–2013 | LEaB; LEa65; IMR; all-cause M | UR; employment-to-population ratio; GDP | correlations; fixed effects regressions |
| 35. | Khouri, Cehlar, Horansky, & Sandorova (2017) | 268 European regions | 2001–2014 | LEaB | IM; % long-term unemployment; population age distribution (<15, 15–64, ≥65yo); #deaths; rate of economic activity; economically active population; employment; employment rate; total fertility; GDP in Euro; GDP in millions of Euro; Creation of Gross Fixed Capital; household income in Euro; household income in millions of Euro; long-term unemployment as % of unemployment; median age of the pop; UR; population density; live births; mean maternal age at birth; gross added value; GDP per capita; GDP as % of EU average; gross birth rate; gross M rate; gross rate of natural movement of population; natural movement of population; gross migration rate; aging index; index of economic dependence of young people; index of economic dependence of old people | fixed & random effects models |
| 36. | Kim & Kim (2017) | 34 European countries | 2000–2012 | LEa60 | GNI; GII; depth of credit information | hierarchical linear regression |
| 37. | Laugesen & Grace (2017) | 22 OECD countries | 1988–1998 | period LE | tobacco consumption; atherogenic-thrombogenic index | correlations; regression |
| 38. | Lenhart (2017) | 24 OECD countries | 1980–2010 | LEaB; overall M | Kaitz wages index; %population above 65yo; %male population; %civilian labor force; GDP; government HE; #hospital beds; public spending; marginal tax rate | fixed effects ordinary least squares regression |
| 39. | Linden & Ray (2017) | 34 OECD countries | 1970–2012 | LEaB | public HE; private HE | dynamic time-series analysis |
| 40. | Marinacci, Demaria, Melis, Borrell, Corman, Dell'Olmo, Rodriguez, & Costa (2017) | 4 European cities | 2000–2011 | all-cause M | Caranci index of socio-economic deprivation; segregation of socio-economically disadvantaged residents | multilevel models |
| 41. | Patton, D., Costich, J. F., & Lidstromer, N. (2017) | 19 OECD countries | 1960–2012 | IMR; post-neonatal MR | job-protected paid parental leave; total fertility rate; female labor force participation; % insured; GDP; HE; low birth weight; family benefits | generalized least-squares regression |

(*Continued*)

**Table 1.** (Continued)

| | author(s) | region | time span | outcome(s) | covariates* | methods |
|---|---|---|---|---|---|---|
| 42. | Richardson, Moon, Pearce, Shortt., & Mitchell (2017) | 274 cities from 27 European countries | 1999–2009 | urban M | GDP | multilevel linear regression models |
| 43. | Tavares (2017) | 28 EU countries | 2005–2012 | IMR | GDP; public HE; UR; % population at risk of poverty, severely materially deprived or living in households with very low work intensity; Gini index; %population with at least lower secondary education; % live births to mothers younger than 20yo; mother's mean age at the first child | robust & panel data regressions |
| 44. | Aguilar-Palacio, Gil-Lacruz, Sanchez-Recio, & Rabanaque (2018) | 14 European countries | 2006–2009 | self-rated health | welfare system typology: Bismarckian, Eastern, and Southern | multilevel models with a logistic function |
| 45. | Blazquez-Fernandez, Cantarero-Prieto, & Pascual-Saez (2018) | 26 European countries | 1995–2014 | LEaB | GDP; Gini coefficient of equalized disposable income; primary school enrollment; total HE; #total hospital beds; S80/S20 income quintile ratio | correlations; panel data models: fixed effects, random effects, feasible generalized least squares |
| 46. | Ferreira, Monteiro, & Manso (2018) | 15 EU countries | 1990–2013 | all-cause M | real (public) social welfare expenditures; real public HE; out-of-pocket HE; GDP; % population >65yo | fixed effects least squares regression |
| 47. | Kolip & Lange (2018) | 28 EU countries | 2015 | LEaB | GII | Pearson correlation coefficients |
| 48. | Korotayev, Khaltourina, Meshcherina, & Zamiatnina (2018) | 40 European countries | 2005; 2010 | MR | recorded & unrecorded alcohol consumption (>15yo); total HE; smoking prevalence (among males); %population 15-64yo consuming opiates; injected drugs prevalence among 15-64yo; fruit and vegetable consumption | ordinary least squares multiple regression |
| 49. | Liobikiene & Bernatoniene (2018) | 27 EU countries | 2014 | self-rated health | GDP | Spearman correlation coefficient |
| 50. | Rajmil, Taylor-Robinson, Gunnlaugsson, Hjern, & Spencer (2018) | 16 EEA countries | 2005–2015 | IM | cyclically adjusted primary balance | longitudinal generalized estimating equations model |
| 51. | Reynolds (2018) | 16 wealthy countries | 1960–2010 | LEaB | HC effort (public HE as % of GDP); pub. sector share (public HE as % of total); GDP; Gini coefficient; % population (> = 25yo) w/ completed tertiary schooling; UR; union density; cigarette consumption (>15yo); net migration; % elderly pop (> = 65yo); total fertility rate; left cabinet | fixed effects models |
| 52. | Reynolds & Avendano (2018) | 20 OECD countries | 1980–2010 | period LE | social spending; GDP; UR; Gini index; population age distribution (<15, 15–64, ≥65yo); government HE | fixed effects models |
| 53. | Ribeiro, Krainski, Carvalho, Launoy, Pornet, & de Pina (2018) | 1911 areas in 5 European countries | 2001–2011 | old-age survival | European Deprivation Index | hierarchical Bayesian spatial models; flexible regression models |
| 54. | Ribeiro, Fraga, & Barros (2018) | 74 cities in 29 European countries | 2013 | all-cause M | residents' global dissatisfaction; % dissatisfied by domains of urban living: physical, social, economic environment, healthcare, and infrastructures/services | generalized linear models (Gaussian) |
| 55. | Tavares (2018) | 28 EU countries | 2013/2014 | self-reported general health status | ICT Development Index; eHealth Index at General Practitioner level; public HE; % population with basic secondary education | ordinary least squares linear regression |
| 56. | Ballester, Robine, Herrmann, & Rodo (2019) | 140 regions in 15 European countries | 2000–2010 | MR | GDP | Pearson correlation coefficients |

(*Continued*)

**Table 1.** (Continued)

| | author(s) | region | time span | outcome(s) | covariates* | methods |
|---|---|---|---|---|---|---|
| 57. | Borisova (2019) | 27 CEE countries | 2005/ 2006 & 2009/ 2010 | subjective health | GDP; Corruption Perception Index; associations membership; trust in society; average length of hospital stay | multi-level analysis using maximum likelihood estimation |
| 58. | Bosakova & Rosicova (2019) | Visegrad countries | 2011–2013 | total M | long-term UR; social exclusion; % population 25-64yo with only lower secondary education | spatial autoregressive regression |
| 59. | Park & Nam (2019) | 27 OECD countries | 1994–2012 | LEaB; MR; IMR; PYLL | GDP; civilian labor force; school LE; UR; wastewater treatment; nitrous oxide (NO) emissions; PM10 emissions; sulfur oxide emissions; tobacco consumption (>15yo); alcohol consumption (>15yo); sugar consumption; calorie intake; vegetable consumption; fat consumption; #physicians per 1000; #medical & social workers per 1000; #hosp. beds per 1000; total HE; measles vaccination rate | fixed effects regression |
| 60. | Rajmil & de Sanmamed (2019) | 15 European countries | 2011–2015 | MR | cyclically adjusted primary balance | longitudinal generalized estimating equations model |

LE(aB;a40;a60;a65) = life expectancy (at birth; at 40 yrs of age; at 60 yrs of age; at 65 yrs of age); M(R) = mortality (rate); IM(R) = infant mortality (rate); CM(R) = child mortality (rate); DALE = disability-adjusted life expectancy; HHDI = health human development index; HLE = healthy life expectancy; PYLL = potential years of life lost; UR = unemployment rate; GDP = gross domestic product; HE = health(care) expenditure; GNI = gross nation income; GII = gender inequality index

* only aggregate level covariates listed and regardless of whether they were treated as main covariates or controls in the particular analysis

The fact that all notable pairwise "connections" are concentrated within a relatively small region of the plot may be interpreted as low overall "connectedness" among the health indicators studied. GDP is the most widely investigated determinant in relation to general population health. Its total number of "connections" is disproportionately high (159) compared to its runner-up–HE (with 113 "connections"), and then subsequently EDU (with 90) and UR (with 86). In fact, all of these determinants could be thought of as outliers, given that none of the remaining factors have a total count of pairings above 52. This decrease in individual determinants' overall "connectedness" can be tracked on the graph via the change of color intensity as we move outwards from the symbolic center of GDP and its closest "co-determinants", to finally reach the other extreme of the ten indicators (welfare regime, household consumption, compulsory school reform, life satisfaction, government revenues, literacy, research expenditure, multiple pregnancy, Cyclically Adjusted Primary Balance, and residents' dissatisfaction; in white) the effects on health of which were only studied in isolation.

Lastly, we point to the few small but stable clusters of covariates encircled by the grey bubbles on Fig 2. These groups of determinants were identified as "close" by both statistical procedures used for the production of the graph (see details in S2 Appendix).

## Statistical methodology

There was great variation in the level of statistical detail reported. Some authors provided too vague a description of their analytical approach, necessitating some inference in this section.

The issue of missing data is a challenging reality in this field of research, but few of the studies under review (12/60) explain how they dealt with it. Among the ones that do, three general approaches to handling missingness can be identified, listed in increasing level of

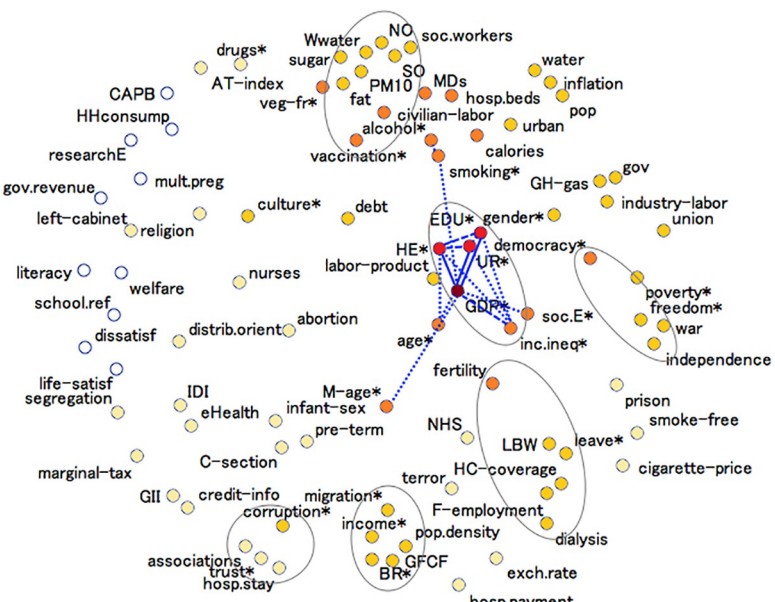

**Fig 2. "Map" of determinants connectedness.** Groups of determinants are marked by asterisks (see S1 Table in S1 Appendix). Diminishing color intensity reflects a decrease in the total number of "connections" for a given determinant. Noteworthy pairwise "connections" are emphasized via lines (solid-dashed-dotted indicates decreasing frequency). Grey contour lines encircle groups of variables that were identified via cluster analysis. Abbreviations: age = population age distribution, associations = membership in associations, AT-index = atherogenic-thrombogenic index, BR = birth rate, CAPB = Cyclically Adjusted Primary Balance, civilian-labor = civilian labor force, C-section = Cesarean delivery rate, credit-info = depth of credit information, dissatisf = residents' dissatisfaction, distrib. orient = distributional orientation, EDU = education, eHealth = eHealth index at GP-level, exch.rate = exchange rate, fat = fat consumption, GDP = gross domestic product, GFCF = Gross Fixed Capital Formation/Creation, GH-gas = greenhouse gas, GII = gender inequality index, gov = governance index, gov.revenue = government revenues, HC-coverage = healthcare coverage, HE = health(care) expenditure, HHconsump = household consumption, hosp.beds = hospital beds, hosp.payment = hospital payment scheme, hosp.stay = length of hospital stay, IDI = ICT development index, inc.ineq = income inequality, industry-labor = industrial labor force, infant-sex = infant sex ratio, labor-product = labor production, LBW = low birth weight, leave = work leave, life-satisf = life satisfaction, M-age = maternal age, marginal-tax = marginal tax rate, MDs = physicians, mult.preg = multiple pregnancy, NHS = Nation Health System, NO = nitrous oxide emissions, PM10 = particulate matter (PM10) emissions, pop = population size, pop.density = population density, pre-term = pre-term birth rate, prison = prison population, researchE = research&development expenditure, school.ref = compulsory schooling reform, smoke-free = smoke-free places, SO = sulfur oxide emissions, soc.E = social expenditure, soc.workers = social workers, sugar = sugar consumption, terror = terrorism, union = union density, UR = unemployment rate, urban = urbanization, veg-fr = vegetable-and-fruit consumption, welfare = welfare regime, Wwater = wastewater treatment.

sophistication: case-wise deletion, i.e., removal of countries from the sample [20, 45, 48, 58, 59], (linear) interpolation [28, 30, 34, 58, 59, 63], and multiple imputation [26, 41, 52].

Correlations, Pearson, Spearman, or unspecified, were the only technique applied with respect to the health outcomes of interest in eight analyses [33, 42–44, 46, 53, 57, 61]. Among the more advanced statistical methods, the family of regression models proved to be, by and large, predominant. Before examining this closer, we note the techniques that were, in a way, "unique" within this selection of studies: meta-analyses were performed (random and fixed effects, respectively) on the reduced form and 2-sample two stage least squares (2SLS) estimations done within countries [39]; difference-in-difference (DiD) analysis was applied in one case [23]; dynamic time-series methods, among which co-integration, impulse-response function (IRF), and panel vector autoregressive (VAR) modeling, were utilized in one study [80]; longitudinal generalized estimating equation (GEE) models were developed on two occasions [70, 78]; hierarchical Bayesian spatial models [51] and special autoregressive regression [62] were also implemented.

Purely cross-sectional data analyses were performed in eight studies [25, 45, 47, 50, 55, 56, 67, 71]. These consisted of linear regression (assumed ordinary least squares (OLS)), generalized least squares (GLS) regression, and multilevel analyses. However, six other studies that used longitudinal data in fact had a cross-sectional design, through which they applied regression at multiple time-points separately [27, 29, 36, 48, 68, 72].

Apart from these "multi-point cross-sectional studies", some other simplistic approaches to longitudinal data analysis were found, involving calculating and regressing 3-year averages of both the response and the predictor variables [54], taking the average of a few data-points (i.e., survey waves) [56] or using difference scores over 10-year [19, 29] or unspecified time intervals [40, 55].

Moving further in the direction of more sensible longitudinal data usage, we turn to the methods widely known among (health) economists as "panel data analysis" or "panel regression". Most often seen were models with fixed effects for country/region and sometimes also time-point (occasionally including a country-specific trend as well), with robust standard errors for the parameter estimates to take into account correlations among clustered observations [20, 21, 24, 28, 30, 32, 34, 37, 38, 41, 52, 59, 60, 63, 66, 69, 73, 79, 81, 82]. The Hausman test [83] was sometimes mentioned as the tool used to decide between fixed and random effects [26, 49, 63, 66, 73, 82]. A few studies considered the latter more appropriate for their particular analyses, with some further specifying that (feasible) GLS estimation was employed [26, 34, 49, 58, 60, 73]. Apart from these two types of models, the first differences method was encountered once as well [31]. Across all, the error terms were sometimes assumed to come from a first-order autoregressive process (AR(1)), i.e., they were allowed to be serially correlated [20, 30, 38, 58–60, 73], and lags of (typically) predictor variables were included in the model specification, too [20, 21, 37, 38, 48, 69, 81]. Lastly, a somewhat different approach to longitudinal data analysis was undertaken in four studies [22, 35, 48, 65] in which multilevel–linear or Poisson–models were developed.

Regardless of the exact techniques used, most studies included in this review presented multiple model applications within their main analysis. None attempted to formally compare models in order to identify the "best", even if goodness-of-fit statistics were occasionally reported. As indicated above, many studies investigated women's and men's health separately [19, 21, 22, 27–29, 31, 33, 35, 36, 38, 39, 45, 50, 51, 64, 65, 69, 82], and covariates were often tested one at a time, including other covariates only incrementally [20, 25, 28, 36, 40, 50, 55, 67, 73]. Furthermore, there were a few instances where analyses within countries were performed as well [32, 39, 51] or where the full time period of interest was divided into a few sub-periods [24, 26, 28, 31]. There were also cases where different statistical techniques were applied in parallel [29, 55, 60, 66, 69, 73, 82], sometimes as a form of sensitivity analysis [24, 26, 30, 58, 73]. However, the most common approach to sensitivity analysis was to re-run models with somewhat different samples [39, 50, 59, 67, 69, 80, 82]. Other strategies included different categorization of variables or adding (more/other) controls [21, 23, 25, 28, 37, 50, 63, 69], using an alternative main covariate measure [59, 82], including lags for predictors or outcomes [28, 30, 58, 63, 65, 79], using weights [24, 67] or alternative data sources [37, 69], or using non-imputed data [41].

## Findings

As the methods and not the findings are the main focus of the current review, and because generic checklists cannot discern the underlying quality in this application field (see also below), we opted to pool all reported findings together, regardless of individual study characteristics or particular outcome(s) used, and speak generally of positive and negative effects on

health. For this summary we have adopted the 0.05-significance level and only considered results from multivariate analyses. Strictly birth-related factors are omitted since these potentially only relate to the group of infant mortality indicators and not to any of the other general population health measures.

Starting with the determinants most often studied, higher GDP levels [21, 26, 27, 29, 30, 32, 43, 48, 52, 58, 60, 66, 67, 73, 79, 81, 82], higher health [21, 37, 47, 49, 52, 58, 59, 68, 72, 82] and social [20, 21, 26, 38, 79] expenditures, higher education [26, 39, 52, 62, 72, 73], lower unemployment [60, 61, 66], and lower income inequality [30, 42, 53, 55, 73] were found to be significantly associated with better population health on a number of occasions. In addition to that, there was also some evidence that democracy [36] and freedom [50], higher work compensation [43, 79], distributional orientation [54], cigarette prices [63], gross national income [22, 72], labor productivity [26], exchange rates [32], marginal tax rates [79], vaccination rates [52], total fertility [59, 66], fruit and vegetable [68], fat [52] and sugar consumption [52], as well as bigger depth of credit information [22] and percentage of civilian labor force [79], longer work leaves [41, 58], more physicians [37, 52, 72], nurses [72], and hospital beds [79, 82], and also membership in associations, perceived corruption and societal trust [48] were beneficial to health. Higher nitrous oxide (NO) levels [52], longer average hospital stay [48], deprivation [51], dissatisfaction with healthcare and the social environment [56], corruption [40, 50], smoking [19, 26, 52, 68], alcohol consumption [26, 52, 68] and illegal drug use [68], poverty [64], higher percentage of industrial workers [26], Gross Fixed Capital creation [66] and older population [38, 66, 79], gender inequality [22], and fertility [26, 66] were detrimental.

It is important to point out that the above-mentioned effects could not be considered stable either across or within studies. Very often, statistical significance of a given covariate fluctuated between the different model specifications tried out within the same study [20, 49, 59, 66, 68, 69, 73, 80, 82], testifying to the importance of control variables and multivariate research (i.e., analyzing multiple independent variables simultaneously) in general. Furthermore, conflicting results were observed even with regards to the "core" determinants given special attention, so to speak, throughout this text. Thus, some studies reported negative effects of health expenditure [32, 82], social expenditure [58], GDP [49, 66], and education [82], and positive effects of income inequality [82] and unemployment [24, 31, 32, 52, 66, 68]. Interestingly, one study [34] differentiated between temporary and long-term effects of GDP and unemployment, alluding to possibly much greater complexity of the association with health. It is also worth noting that some gender differences were found, with determinants being more influential for males than for females, or only having statistically significant effects for male health [19, 21, 28, 34, 36, 37, 39, 64, 65, 69].

## Discussion

The purpose of this scoping review was to examine recent quantitative work on the topic of multi-country analyses of determinants of population health in high-income countries.

Measuring population health via relatively simple mortality-based indicators still seems to be the state of the art. What is more, these indicators are routinely considered one at a time, instead of, for example, employing existing statistical procedures to devise a more general, composite, index of population health, or using some of the established indices, such as disability-adjusted life expectancy (DALE) or quality-adjusted life expectancy (QALE). Although strong arguments for their wider use were already voiced decades ago [84], such summary measures surface only rarely in this research field.

On a related note, the greater data availability and accessibility that we enjoy today does not automatically equate to data quality. Nonetheless, this is routinely assumed in aggregate level

studies. We almost never encountered a discussion on the topic. The non-mundane issue of data missingness, too, goes largely underappreciated. With all recent methodological advancements in this area [85–88], there is no excuse for ignorance; and still, too few of the reviewed studies tackled the matter in any adequate fashion.

Much optimism can be gained considering the abundance of different determinants that have attracted researchers' attention in relation to population health. We took on a visual approach with regards to these determinants and presented a graph that links spatial distances between determinants with frequencies of being studies together. To facilitate interpretation, we grouped some variables, which resulted in some loss of finer detail. Nevertheless, the graph is helpful in exemplifying how many effects continue to be studied in a very limited context, if any. Since in reality no factor acts in isolation, this oversimplification practice threatens to render the whole exercise meaningless from the outset. The importance of multivariate analysis cannot be stressed enough. While there is no "best method" to be recommended and appropriate techniques vary according to the specifics of the research question and the characteristics of the data at hand [89–93], in the future, in addition to abandoning simplistic univariate approaches, we hope to see a shift from the currently dominating fixed effects to the more flexible random/mixed effects models [94], as well as wider application of more sophisticated methods, such as principle component regression, partial least squares, covariance structure models (e.g., structural equations), canonical correlations, time-series, and generalized estimating equations.

Finally, there are some limitations of the current scoping review. We searched the two main databases for published research in medical and non-medical sciences (PubMed and Web of Science) since 2013, thus potentially excluding publications and reports that are not indexed in these databases, as well as older indexed publications. These choices were guided by our interest in the most recent (i.e., the current state-of-the-art) and arguably the highest-quality research (i.e., peer-reviewed articles, primarily in indexed non-predatory journals). Furthermore, despite holding a critical stance with regards to some aspects of how determinants-of-health research is currently conducted, we opted out of formally assessing the quality of the individual studies included. The reason for that is two-fold. On the one hand, we are unaware of the existence of a formal and standard tool for quality assessment of ecological designs. And on the other, we consider trying to score the quality of these diverse studies (in terms of regional setting, specific topic, outcome indices, and methodology) undesirable and misleading, particularly since we would sometimes have been rating the quality of only a (small) part of the original studies—the part that was relevant to our review's goal.

Our aim was to investigate the current state of research on the very broad and general topic of population health, specifically, the way it has been examined in a multi-country context. We learned that data treatment and analytical approach were, in the majority of these recent studies, ill-equipped or insufficiently transparent to provide clarity regarding the underlying mechanisms of population health in high-income countries. Whether due to methodological shortcomings or the inherent complexity of the topic, research so far fails to provide any definitive answers. It is our sincere belief that with the application of more advanced analytical techniques this continuous quest could come to fruition sooner.

## Supporting information

**S1 Checklist. Preferred reporting items for systematic reviews and meta-analyses extension for scoping reviews (PRISMA-ScR) checklist.**
(DOCX)

**S1 Appendix.**
(DOCX)

**S2 Appendix.**
(DOCX)

## Author Contributions

**Conceptualization:** Vladimira Varbanova, Philippe Beutels.

**Data curation:** Vladimira Varbanova, Philippe Beutels.

**Formal analysis:** Vladimira Varbanova.

**Funding acquisition:** Philippe Beutels.

**Investigation:** Vladimira Varbanova.

**Methodology:** Vladimira Varbanova.

**Project administration:** Vladimira Varbanova, Philippe Beutels.

**Resources:** Philippe Beutels.

**Software:** Vladimira Varbanova.

**Supervision:** Philippe Beutels.

**Validation:** Philippe Beutels.

**Visualization:** Vladimira Varbanova, Philippe Beutels.

**Writing – original draft:** Vladimira Varbanova.

**Writing – review & editing:** Vladimira Varbanova, Philippe Beutels.

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
