## [Decision Letter · Decision Letter 0]

19 Dec 2019

PONE-D-19-19419

Recent quantitative research on determinants of health in high income countries: A scoping review

PLOS ONE

Dear Varbanova,

Thank you for submitting your manuscript to PLOS ONE. After careful consideration, we feel that it has merit but does not fully meet PLOS ONE’s publication criteria as it currently stands. Therefore, we invite you to submit a revised version of the manuscript that addresses the points raised during the review process.

We would appreciate receiving your revised manuscript by 26 January 2020. To enhance the reproducibility of your results, we recommend that if applicable you deposit your laboratory protocols in protocols.io, where a protocol can be assigned its own identifier (DOI) such that it can be cited independently in the future. For instructions see: http://journals.plos.org/plosone/s/submission-guidelines#loc-laboratory-protocols

We look forward to receiving your revised manuscript.

Kind regards,

Russell Kabir, PhD

Academic Editor

PLOS ONE

Journal Requirements:

1.

Reviewers' comments:

Reviewer's Responses to Questions

**Comments to the Author**

1. Is the manuscript technically sound, and do the data support the conclusions?

Reviewer #1: Partly

Reviewer #2: Yes

2. Has the statistical analysis been performed appropriately and rigorously? 

Reviewer #1: I Don't Know

Reviewer #2: N/A

3. Have the authors made all data underlying the findings in their manuscript fully available?

Reviewer #1: Yes

Reviewer #2: Yes

4. Is the manuscript presented in an intelligible fashion and written in standard English?

Reviewer #1: Yes

Reviewer #2: Yes

5. Review Comments to the Author

Reviewer #1: Thank you for submitting this scoping review. I have reviewed this paper in line with the PRISMA statement for scoping reviews.

Introduction - information is provided about previous approaches to investigating population health. Examples would be useful so that the reader does not have to refer to the reference list for an example of 'just a single health predictor'. The focus on high income countries needs defending here.

Methods

Search - please be specific the use of etc. is vague e.g. line 91. Need a defence of the choice of the two databases (arguably this is narrow scope for a scoping review). It would be better if the critieria are written under the headings of inclusion and exclusion. Need reference support for choices made e.g. c) d) and e). Please include years considered and language. In line with the scoping review statement something is needed for items 12 and 13 - critical appraisal and synthesis of results. The income dichotomy used in study selection needs defining and explaining here. The statistical pooling employed needs detailing and defending here. Line 152, a 0 is used instead of O.

Results

p 10 line 179 - this is confusing - if these papers are not within the reviews scope then why are they included? Database URLs are needed - line 240. Line 385 - the five studies referred to need references. Line 393 - what makes these relationships 'weaker'. p27 the statistical pooling is not detailed in the methods it is unclear what is done here and if this was planned apriori or decided post hoc (because there is not protocol published). Why is this pooling done given the comment on line 516 (about the lack of stability of the findings)?

Discussion - this section is short. Why wasn't the rationale of each paper extracted in relation to the defence of the choice of methods (for e.g. for population health indicators ). The study authors may have reasons (not captured here) for choosing this approach? Little argument about the pro's and con's of each method is presented. For each of the data items extracted an argument is needed about the choices available, which method is deemed 'best' and why. This section is largely limitations - here the use of databases is defended based on focusing on 'high quality research' why is this so? i.e. why is only high quality research published in these databases? how were judgements about quality made?

Reviewer #2: The authors reviewed the recent quantitative work on the analyses of determinants of population health in high

income countries and mapped a wide range of literature. The research gap and the message that the authors want to convey to statistical modelers and policy makers are not very clear. In addition, the authors are advised to address the following points:

1. Different statistical methods are using in different studies. Efficiency and drawbacks of the methods should be briefly discussed.

2. In line 560 "The importance of multivariate analysis cannot be stressed enough". The reason behind this suggestion is not clear.

3. Some of the methods used to handle missingness (lines 418-423) are known to be biased. A brief comment on this is recommended.

4. Figures on health determinants are known to vary depending on how the data is obtained and methods used to analyze it. This might introduce difference among the results. Please discuss it briefly.

5. Issues related to data quality are well addressed. What about integration of data coming from various sources.

6. In this study, literature published between 2013 and 2019 are reviewed. Changes in policy, changes in definition of the outcome variables might limit the methods and affect the analysis. Has this been assessed? If not, potential impact of change in policy or definition of variables should be amended.

7. Use of more sophisticated methodology is commended to analyze such datasets. This is very general. Better to be specific.

6. PLOS authors have the option to publish the peer review history of their article (what does this mean?). If published, this will include your full peer review and any attached files.

Reviewer #1: Yes: Dr Leica Claydon-Mueller

Reviewer #2: No

---

## [Author Response · Author response to Decision Letter 0]

24 Jan 2020

Reviewer #1

Introduction

“Examples would be useful so that the reader does not have to refer to the reference list for an example of 'just a single health predictor'.”

Implemented in lines 61-64: “In some cases, the impact of one or more interventions is at the core of the review (1-5), while in others it is the relationship between health and just one particular predictor, e.g., income inequality, access to healthcare, government mechanisms (6-11).”

“The focus on high income countries needs defending here.”

Implemented in lines 74-86: “The goal of this scoping review was to document the state of the art in the recent published literature on determinants of population health, with a particular focus on the types of determinants selected as well as the methodology used. In doing so, we also report the main characteristics of the results these studies found. The materials collected in this review were meant to inform our (and potentially other researchers’) future analyses on this topic. Since the production of health is subject to the law of diminishing marginal returns, we focused only on those studies that included countries where a high standard of wealth has been achieved for some time, i.e., high-income countries belonging to the Organisastion for Economic Co-operation and Development (OECD) or Europe. Adding similar reviews for other country income groups is of limited interest to the research we plan to do in this area and would have made the current paper too long.” 

Methods

“please be specific the use of etc. is vague e.g. line 91”

In order to understand this comment, we did a search within the manusript and found we had used “etc.” twice in the entire text. 

In lines 48-49: “In addition, attempts to measure health can be made at the individual level or some aggregate, such as neighborhood, regional, national, etc.” This sentence was modified to: “In addition, attempts to measure health can be made at the individual level or some aggregate, such as a neighborhood, a region or a country.”

In lines 139-140: “…the analysis involved at least two countries or at least two regions, cities, etc. in at least two different countries;” This sentence was modified to: “…the analysis involved at least two countries or at least two regions or cities (or another aggregate level) in at least two different countries;”

“Need a defence of the choice of the two databases (arguably this is narrow scope for a scoping review).”

Pubmed and Web of Science (which together cover all areas of science to a larger extent than Scopus) are the most important reproducible human-curated databases of scientific literature covering journals and books. Google scholar is non-human curated and contains much predatory journal and other unscientific content, which – given the general search terms we have had to use – would have lead to a huge additional triage burden, for highly uncertain gains in our review. It is highly unlikely that state-of-the-science methods used in contributions in Embase and Scopus over the last 5-6 years would not have been findable in Pubmed and Web of Science. Nonetheless, we mention the choice of these two databases as a limitation in the discussion as follows (lines 586-592): “We searched the two main databases for published research in medical and non-medical sciences (Pubmed and Web of Science) since 2013, thus potentially excluding publications and reports that are not indexed in these databases, as well as older indexed publications. These choices were guided by our interest in the most recent (i.e., the current state-of-the-art) and arguably the highest-quality research (i.e., peer-reviewed articles, primarily in indexed non-predatory journals).”

“Need reference support for choices made e.g. c) d) and e) [of eligibility criteria].”

We were interested in cross-country comparisons and therefore needed at least two different countries to be considered in a particular study (point c). As per our choice to exclude studies that used differentiated health outcomes or cause-specific mortality rates (points d) and e), respectively), this was dictated by our desire to have a broad overview on the topic of general population health determinants. We believe analyses on this topic present specific difficulties on which we were hoping to learn from the literature. The excluded papers utilizing population outcomes represented by health inequality and specific causes of death are themselves part of a huge body of literature that would require separate literature reviews to study appropriately. 

“Please include years considered and language.”

We refer to lines 125-127, where we stated: “Both searches were performed on the 16th of July, 2019, without any language restrictions, and with a start date set to the 1st of January, 2013, as we were interested in the latest developments in this area of research.”

 “In line with the scoping review statement something is needed for items 12 and 13 - critical appraisal and synthesis of results.” 

Critical appraisal is optional for a scoping review (12) and we chose not to complete it because of the reasons stated in lines 592-602: “Furthermore, despite holding a critical stand with regards to some aspects of the way determinants-of-health research is currently being conducted, we opted out of formally assessing the quality of the individual studies included. The reason for that is two-fold: on the one hand, we are unaware of the existence of a formal and standard tool for quality assessment of ecological designs; and on the other, we consider trying to score the quality of these diverse studies (in terms of regional setting, specific topic, outcome indices, and methodology) undesirable and misleading. All the more so since we would sometimes have been rating the quality of only a (small) part of the original studies - the part that was relevant to our review’s goal.”

Clarification on the synthesis of results has been added in the text in lines 147-153: “The following pieces of information were extracted in an Excel table from the full text of each eligible study (primarily by VV, consulting with PB in case of doubt): health outcome(s), determinants, statistical methodology, level of analysis, results, type of data, data sources, time period, countries. The evidence is synthesized according to these extracted data (often directly reflected in the section headings), using a narrative form accompanied by a “summary-of-findings” table and a graph.” 

“The income dichotomy used in study selection needs defining and explaining here.”

Implemented in lines 167-177: “This group was further dichotomized according to level of economic development of the countries or regions under study, using membership of the OECD or Europe as a reference “cut-off” point. Sixty papers were judged to include high-income countries, and the remaining 142 included either low- or middle-income countries or a mix of both these levels of development. The rest of this report outlines findings in relation to high-income countries only, reflecting our own primary research interests. Nonetheless, we chose to report our search yield for the other income groups for two reasons: First, to gauge the relative interest in applied published research for these different income levels; and second, to enable other researchers with a focus on determinants of health in other countries to use the extraction we made here.” 

“The statistical pooling employed needs detailing and defending here.”

This was not a statistical pooling in the strict sense but rather an overview of the findings of the selected studies. Details are outlined in lines 501-509: “As the methods and not the findings are the main focus of the current review, and as generic checklists cannot discern the underlying quality in this application field (see also below), we opted to pool all reported findings together, regardless of individual study characteristics or particular outcome(s) used, and quite generally speak of positive and negative effects on health. Please note that for this summary we have adopted the 0.05-significance level and only considered results from multivariate analyses. Strictly birth-related factors are omitted since these potentially only relate to the group of infant mortality indicators and not to any of the other general population health measures.”

“Line 152, a 0 is used instead of O.”

The line in question (now 160) reads: “effectiveness, with theoretical or non-health related issues, animals or plants. Of “ The second to last symbol on that line is indeed the letter O and not the number 0. 

Results

“p 10 line 179 - this is confusing - if these papers are not within the reviews scope then why are they included?”

The sentence in question read: “While the majority of studies under review here focused on a single health indicator, 23 out of the 60 studies made use of multiple outcomes, but these were always considered one at a time, and sometimes not all of these fell within the scope of our review.” The word “these” as seen twice in the second half of the sentence referred to “multiple outcomes” that immediately precedes and not to “studies” mentioned earlier. To reiterate, papers were included if at least one of the considered health outcomes fell within the scope of the review, even if other considered health outcomes in that same study fell outside of our scope. To avoid all confusion for future readers, we have emphasised this in the said sentence as: “While the majority of studies under review here focused on a single health indicator, 23 out of the 60 studies made use of multiple outcomes, but these outcomes were always considered one at a time, and sometimes not all of them fell within the scope of our review.” (lines 192-195).

“Database URLs are needed - line 240.”

Implemented for the five most widely used databases in lines 253-257: “The accessible reference databases of the OECD (https://www.oecd.org/), WHO (https://www.who.int/), World Bank (https://www.worldbank.org/), United Nations (https://www.un.org/en/), and Eurostat (https://ec.europa.eu/eurostat) were among the top choices.” We find including URL’s for all sources that we make mention of unnecessary, all the more so as the list provided in the text is not exhaustive and very few original papers were transparent in reporting the URL’s they used to access the databases. 

“Line 385 - the five studies referred to need references.”

Implemented in lines 398-400 (please note that there are 6 studies in total, of which we need to cite five in both instances): “In reality however, smoking was considered together with alcohol consumption (13-17) in just as many studies as it was with GDP (13-15, 17, 18), five.”

“Line 393 - what makes these relationships 'weaker'.”

The fact that these factors were studied together less frequently (than the pairs that were just mentioned in the previous two sentences) – lines 401-409: “Solid lines connect GDP with health expenditure (HE), unemployment rate (UR), and education (EDU), indicating that the effect of GDP on health, taking into account the effects of the other three determinants as well, was evaluated in between 12 to 16 studies of the 60 included in this review. Tracing the dashed lines, we can also tell that GDP appeared jointly with income inequality and HE together with either EDU or UR in anywhere between 8 to 10 of our selected studies. Finally, some weaker but still worth-mentioning “connections” between variables are displayed as well via the dotted lines.”

“p27 the statistical pooling is not detailed in the methods it is unclear what is done here and if this was planned apriori or decided post hoc (because there is not protocol published). Why is this pooling done given the comment on line 516 (about the lack of stability of the findings)?”

We consider it important to give a rough indication of the diversity of results of the included studies, though - as we stated – we have found that no conclusions with regards to any health determinant could be drawn (this review was not intended to include a meta-analysis, and it should be clear to the reader that the literature on this does not allow conducting a meta-analysis). As outlined in lines 501-509 re-printed below, the effects mentioned in the text were all found to be statistically significant at the 0.05-level in at least one study that did not do that via a univariate analysis (i.e., there were 2 or more determinants considered at the same time) and also were not strictly birth-related. How we determined “good”/”bad” for health was based on the particular health outcome being desirable or avoidable. For example, determinant-A’s positive association with life expectancy and determinant-B’s negative association with overall mortality were both interpreted as positive effects on health. (“As the methods and not the findings are the main focus of the current review, and as generic checklists cannot discern the underlying quality in this application field (see also below), we opted to pool all reported findings together, regardless of individual study characteristics or particular outcome(s) used, and quite generally speak of positive and negative effects on health. Please note that for this summary we have adopted the 0.05-significance level and only considered results from multivariate analyses. Strictly birth-related factors are omitted since these potentially only relate to the group of infant mortality indicators and not to any of the other general population health measures.”)

Discussion

“Why wasn't the rationale of each paper extracted in relation to the defence of the choice of methods (for e.g. for population health indicators ). The study authors may have reasons (not captured here) for choosing this approach? Little argument about the pro's and con's of each method is presented.”

Critical appraisal was not performed partly in recognition of the fact that researchers had their own rationale for conducting a particular study (lines 592-602): “Furthermore, despite holding a critical stand with regards to some aspects of the way determinants-of-health research is currently being conducted, we opted out of formally assessing the quality of the individual studies included. The reason for that is two-fold: on the one hand, we are unaware of the existence of a formal and standard tool for quality assessment of ecological designs; and on the other, we consider trying to score the quality of these diverse studies (in terms of regional setting, specific topic, outcome indices, and methodology) undesirable and misleading. All the more so since we would sometimes have been rating the quality of only a (small) part of the original studies - the part that was relevant to our review’s goal.” We furthermore allude to this issue by pointing out that papers often had their own specific focus (lines 313-317): “It was often the case that research had a more particular focus. Among others, minimum wages (19), hospital payment schemes (20), cigarette prices (21), social expenditure (22), residents’ dissatisfaction (23), income inequality (24, 25), and work leave (26, 27) took center stage.” Please see also our response to the next comment. 

“For each of the data items extracted an argument is needed about the choices available, which method is deemed 'best' and why.”

We deem this suggestion unfeasible because: 1) we, ourselves, are likely not aware of all possible and available choices in terms of data sources, data granularity, and determinants in a given set of countries or regions; 2) some choices are mostly dictated by specific research interest (e.g., countries to include, time period to consider, level of analysis); 3) there is no one “best” approach, especially when it comes to statistical methodology (though some methods are clearly superior to others and we have stated as much in the text - e.g., line 436: “listed in increasing level of sophistication”, line 442: “Among the more advanced statistical methods”, lines 459-460: “some other simplistic approaches to longitudinal data analysis were found”, line 464: “Moving further in the direction of more sensible longitudinal data usage”). 

“… the use of databases is defended based on focusing on 'high quality research' why is this so? i.e. why is only high quality research published in these databases? how were judgements about quality made?” 

We wrote “…arguably the highest-quality research” (line 591). The word “arguably” should not be overlooked. Both Web of Science and Pubmed remain to date internationally the standard databases through which academic research is judged in applications for research funding and academic career development in sciences in general, and health sciences in particular. One would indeed expect that the highest quality research will eventually be published in a peer reviewed journal indexed in these databases. Clearly, this does not mean that research that has never been offered to an indexed peer-reviewed journal or that research offered to a non-indexed journal is by definition of low quality. By “quality” we mean using state-of-the-science data and methods for the research question at hand, in a transparent way. A general aside about different definitions of “quality” in research work, as well as the pro’s and con’s of different research indexing systems is beyond the scope of this paper. 

 

Reviewer #2

“The research gap and the message that the authors want to convey to statistical modelers and policy makers are not very clear.”

The major gap in research on determinants of overall population health that we identified through this review was a general lack of sophisticated approaches to both statistical analysis and data preparation (namely, missing data handling). Apart from the commonly seen under-reporting on methodology, a sizeable part of the body of research we perused focused on only a small number of determinants, all too often tested one at a time. Given the high complexity of the health production process, research should strive to reflect that reality as much as possible by (simultaneously) taking into account a large(r) number of potential health determinants while making use of more refined analytical techniques. Only this way would we be able to reliably explain observed between-country differences in population health levels and manage to build a solid foundation of evidence to inform future public health policy. 

We hope that the changes and clarifications outlined below address this concern further.

 “1. Different statistical methods are using in different studies. Efficiency and drawbacks of the methods should be briefly discussed.”

We found it impractical to attempt to critically discuss specific statistical approaches when so many of them were encountered. With the aim of providing an overview of how research on the very broad topic of determinants of population health has been conducted, we chose to stay descriptive, nevertheless suggesting on numerous ocassions that methods used were not necessarily optimal (e.g., line 436: “listed in increasing level of sophistication”, line 442: “Among the more advanced statistical methods”, lines 459-460: “some other simplistic approaches to longitudinal data analysis were found”, line 464: “Moving further in the direction of more sensible longitudinal data usage”). Furthermore, in the Discussion section we have expanded the following sentence (lines 576-585), referring to some useful textbooks as well: “While there is no “best method” to be recommended and appropriate techniques vary according to the specifics of the research question and the characteristics of the data at hand (28-32), in the future, in addition to abandoning simplistic univariate approaches, we can only hope to see a discernible shift from the currently dominating fixed effects to the more flexible random/mixed effects models (33), and further on to wider application of more sophisticated methods, such as principle component regression, partial least squares, covariance structure models (e.g., structural equations), canonical correlations, time-series and generalized estimating equations.” Commenting on the pro’s and con’s of each of these suggested methods (or the ones actually encountered in our selection of papers) would have made the review veer in the direction of a textbook on statistics, which was far from our intent. 

“2. In line 560 "The importance of multivariate analysis cannot be stressed enough". The reason behind this suggestion is not clear.”

The reason for this statement perhaps transpires better in lines 531-541, in the context of unstable and contradictory results (i.e., estimated statistical effects are known to be biased when important variables are not taken into account; it is thus important to include control variables in the analysis): “We find it imperative to point out that the above-mentioned effects could not be considered stable either across or within studies. Very often, statistical significance of a given covariate fluctuated between the different model specifications tried out within the same study (16, 18, 22, 25, 34-38), testifying to the importance of control variables and multivariate research (i.e., analyzing multiple independent variables simultaneously) in general. Furthermore, conflicting results were observed even with regards to the “core” determinants given special attention, so to speak, throughout this text. Thus, some studies reported negative effects of health expenditure (38, 39), social expenditure (27), GDP (34, 37), and education (38), and positive effects of income inequality (38) and unemployment (16, 17, 37, 39-41).” 

“3. Some of the methods used to handle missingness (lines 418-423) are known to be biased. A brief comment on this is recommended.”

Indeed, bias can be introduced in the results when the assumptions underlying a specific missing data handling method are not met. Without knowing the nature of the missing data mechanism operating in a particular dataset, it is hard to pass judgement whether the steps taken to tackle the issue were the most appropriate. While we, ourselves, are advocates for multiple imputation, we are aware that under certain circumstances (i.e., data is missing completely at random) simple case-wise deletion is not only acceptable but also more efficient than more advanced methods (42). True to our decision to refrain from critiquing individual statistical approaches because 1) we were not intimately familiar with the datasets used; and 2) compare-and-contrast approach to statistical techniques falls beyond the scope of our review, we chose to just acknowledge potential disadvantages in employed missing data “remedies” by including the phrase “in increasing level of sophistication” (line 436) prior to listing the ones encountered in the selected papers. 

“4. Figures on health determinants are known to vary depending on how the data is obtained and methods used to analyze it. This might introduce difference among the results. Please discuss it briefly.”

If “figures” here refers to effects of health determinants, this is indeed what we observed and, correspondingly, tried to emphasize in the text in lines 531-541: “We find it imperative to point out that the above-mentioned effects could not be considered stable either across or within studies. Very often, statistical significance of a given covariate fluctuated between the different model specifications tried out within the same study (16, 18, 22, 25, 34-38), testifying to the importance of control variables and multivariate research (i.e., analyzing multiple independent variables simultaneously) in general. Furthermore, conflicting results were observed even with regards to the “core” determinants given special attention, so to speak, throughout this text. Thus, some studies reported negative effects of health expenditure (38, 39), social expenditure (27), GDP (34, 37), and education (38), and positive effects of income inequality (38) and unemployment (16, 17, 37, 39-41).” … and in lines 608-609: “Whether due to methodological shortcomings or to the inherent complexity of the topic, research so far fails to provide any definitive answers.” 

“5. Issues related to data quality are well addressed. What about integration of data coming from various sources.”

None of the papers in the review reported problems with integrating data from different sources (also not in relation to different sources producing different values for the same variable in the same time period). If this would be problematic to the extent that it makes some of the data unreliable or contradictory, this would seem to represent another form of poor data quality. 

“6. In this study, literature published between 2013 and 2019 are reviewed. Changes in policy, changes in definition of the outcome variables might limit the methods and affect the analysis. Has this been assessed? If not, potential impact of change in policy or definition of variables should be amended.”

We are not aware of any changes in policy that would directly affect how this particular type of research is being conducted, especially in a time span of 5-6 years. 

Definitions of outome variables related to the topic have been stable over time. However, we did explicitly point out to one discrepancy with regards to the index of child mortality in lines 203-206: “Child mortality has conventionally been defined as mortality within the first 5 years of life, thus often also called “under-5 mortality”. Nonetheless, Pritchard & Wallace used the term “child mortality” to denote deaths of children younger than 14 years (43).” Given the fact that we could provide only a very general overview of the findings of the included studies and our goal was not to arrive at a definitive answer regarding a particular determinant’s effect on child mortality (as it would have been in a meta-analysis), this discrepancy was deemed immaterial. Furthermore, we also pointed out that two studies did not specify the type of life expectancy index they used (lines 184-185): “In two papers, the age-specificity of life expectancy (be it at birth or another age) was not stated (14, 15).” While clearly an ommision on the researchers’ part, for the same reasons already expressed above, this issue was not considered important for our work. 

“7. Use of more sophisticated methodology is commended to analyze such datasets. This is very general. Better to be specific.”

Indeed, in addition to emphasising the need for multivariate instead of univariate analysis, we give specific examples (without claiming to be exhaustive) in lines 576-585: “While there is no “best method” to be recommended and appropriate techniques vary according to the specifics of the research question and the characteristics of the data at hand (28-32), in the future, in addition to abandoning simplistic univariate approaches, we can only hope to see a discernible shift from the currently dominating fixed effects to the more flexible random/mixed effects models (33), and further on to wider application of more sophisticated methods, such as principle component regression, partial least squares, covariance structure models (e.g., structural equations), canonical correlations, time-series and generalized estimating equations.”

 

REFERENCES

1. Mulreany JP, Calikoglu S, Ruiz S, Sapsin JW. Water privatization and public health in Latin America. Rev Panam Salud Publ. 2006;19(1):23-32.

2. Williams DR, Costa MV, Odunlami AO, Mohammed SA. Moving Upstream: How Interventions That Address the Social Determinants of Health Can Improve Health and Reduce Disparities. J Public Health Man. 2008:S8-S17.

3. Bambra C, Gibson M, Sowden A, Wright K, Whitehead M, Petticrew M. Tackling the wider social determinants of health and health inequalities: evidence from systematic reviews. J Epidemiol Community Health. 2010;64(4):284-91.

4. Newman L, Baum F, Javanparast S, O'Rourke K, Carlon L. Addressing social determinants of health inequities through settings: a rapid review. Health Promot Int. 2015;30:126-43.

5. Pega F, Liu SY, Walter S, Pabayo R, Saith R, Lhachimi SK. Unconditional cash transfers for reducing poverty and vulnerabilities: effect on use of health services and health outcomes in low- and middle-income countries. Cochrane Db Syst Rev. 2017(11).

6. Lynch J, Smith GD, Harper S, Hillemeier M, Ross N, Kaplan GA, et al. Is income inequality a determinant of population health? Part 1. A systematic review. Milbank Q. 2004;82(1):5-99.

7. Houweling TA, Kunst AE. Socio-economic inequalities in childhood mortality in low- and middle-income countries: a review of the international evidence. Br Med Bull. 2010;93:7-26.

8. Rutherford ME, Mulholland K, Hill PC. How access to health care relates to under-five mortality in sub-Saharan Africa: systematic review. Trop Med Int Health. 2010;15(5):508-19.

9. Ciccone DK, Vian T, Maurer L, Bradley EH. Linking governance mechanisms to health outcomes: A review of the literature in low- and middle-income countries. Soc Sci Med. 2014;117:86-95.

10. Kelly C, Hulme C, Farragher T, Clarke G. Are differences in travel time or distance to healthcare for adults in global north countries associated with an impact on health outcomes? A systematic review. Bmj Open. 2016;6(11).

11. Parmar D, Stavropoulou C, Ioannidis JPA. Health outcomes during the 2008 financial crisis in Europe: systematic literature review. Bmj-Brit Med J. 2016;354.

12. Tricco AC, Lillie E, Zarin W, O'Brien KK, Colquhoun H, Levac D, et al. PRISMA Extension for Scoping Reviews (PRISMA-ScR): Checklist and Explanation. Ann Intern Med. 2018;169(7):467-+.

13. Barthold D, Nandi A, Rodriguez JMM, Heymann J. Analyzing Whether Countries Are Equally Efficient at Improving Longevity for Men and Women. Am J Public Health. 2014;104(11):2163-9.

14. Mackenbach JP. Cultural values and population health: a quantitative analysis of variations in cultural values, health behaviours and health outcomes among 42 European countries. Health Place. 2014;28:116-32.

15. Zare H, Gaskin DJ, Anderson G. Variations in life expectancy in Organization for Economic Co-operation and Development countries-1985-2010. Scand J Public Healt. 2015;43(8):786-95.

16. Korotayev A, Khaltourina D, Meshcherina K, Zamiatnina E. Distilled Spirits Overconsumption as the Most Important Factor of Excessive Adult Male Mortality in Europe. Alcohol Alcoholism. 2018;53(6):742-52.

17. Park MB, Nam EW. National Level Social Determinants of Health and Outcomes: Longitudinal Analysis of 27 Industrialized Countries. Sage Open. 2019;9(2).

18. Reynolds MM. Health Care Public Sector Share and the US Life Expectancy Lag: A Country-level Longitudinal Study. Int J Health Serv. 2018;48(2):328-48.

19. Lenhart O. The impact of minimum wages on population health: evidence from 24 OECD countries. Eur J Health Econ. 2017;18(8):1031-9.

20. Wubulihasimu P, Brouwer W, van Baal P. The Impact of Hospital Payment Schemes on Healthcare and Mortality: Evidence from Hospital Payment Reforms in OECD Countries. Health Econ. 2016;25(8):1005-19.

21. Filippidis FT, Laverty AA, Hone T, Been JV, Millett C. Association of Cigarette Price Differentials With Infant Mortality in 23 European Union Countries. Jama Pediatr. 2017;171(11):1100-6.

22. Reynolds MM, Avendano M. Social Policy Expenditures and Life Expectancy in High-Income Countries. Am J Prev Med. 2018;54(1):72-9.

23. Ribeiro AI, Fraga S, Barros H. Residents' Dissatisfaction and All-Cause Mortality. Evidence from 74 European Cities. Front Psychol. 2018;8.

24. Torre R, Myrskyla M. Income inequality and population health: An analysis of panel data for 21 developed countries, 1975-2006. Pop Stud-J Demog. 2014;68(1):1-13.

25. Hu YN, van Lenthe FJ, Mackenbach JP. Income inequality, life expectancy and cause-specific mortality in 43 European countries, 1987-2008: a fixed effects study. Eur J Epidemiol. 2015;30(8):615-25.

26. Shim J. Family leave policy and child mortality: Evidence from 19 OECD countries from 1969 to 2010. Int J Soc Welf. 2016;25(3):215-21.

27. Patton D, Costich JF, Lidstromer N. Paid Parental Leave Policies and Infant Mortality Rates in OECD Countries: Policy Implications for the United States. World Med Health Pol. 2017;9(1):6-23.

28. Wolfgang Karl Härdle LS. Applied Multivariate Statistical Analysis. Berlin, Heidelberg: Springer; 2015.

29. Agresti A. Foundations of Linear and Generalized Linear Models. Hoboken, New Jersey: John Wiley & Sons Inc.; 2015.

30. Garrett Fitzmaurice MD, Geert Verbeke, Geert Molenberghs. Longitudinal Data Analysis. New York: Chapman and Hall/CRC; 2008.

31. A. H. Leyland (Editor) HGE. Multilevel Modelling of Health Statistics: John Wiley & Sons Inc; 2001.

32. Shayle R. Searle GC, Charles E. McCulloch. Variance Components: John Wiley & Sons, Inc.; 1992.

33. Bell A, Jones K. Explaining Fixed Effects: Random Effects Modeling of Time-Series Cross-Sectional and Panel Data. Polit Sci Res Meth. 2015;3(1):133-53.

34. Lopez-Casasnovas G, Soley-Bori M. The Socioeconomic Determinants of Health: Economic Growth and Health in the OECD Countries during the Last Three Decades. Int J Env Res Pub He. 2014;11(1):815-29.

35. Linden M, Ray D. Life expectancy effects of public and private health expenditures in OECD countries 1970-2012: Panel time series approach. Econ Anal Policy. 2017;56:101-13.

36. Tavares AI. Infant mortality in Europe, socio-economic determinants based on aggregate data. Appl Econ Lett. 2017;24(21):1588-96.

37. Khouri S, Cehlar M, Horansky K, Sandorova K. Expected Life Expectancy and Its Determinants in Selected European Countries. Transform Bus Econ. 2017;16(2b):638-55.

38. Blazquez-Fernandez C, Cantarero-Prieto D, Pascual-Saez M. Does Rising Income Inequality Reduce Life Expectancy? New Evidence for 26 European Countries (1995-2014). Global Econ Rev. 2018;47(4):464-79.

39. Blazquez-Fernandez C, Cantarero-Prieto D, Pascual-Saez M. Health expenditure and socio-economic determinants of life expectancy in the OECD Asia/Pacific area countries. Appl Econ Lett. 2017;24(3):167-9.

40. Bartoll X, Mari-Dell'Olmo M. Patterns of life expectancy before and during economic recession, 2003-12: a European regions panel approach. Eur J Public Health. 2016;26(5):783-8.

41. Granados JAT, Ionides EL. Population health and the economy: Mortality and the Great Recession in Europe. Health Econ. 2017;26(12):E219-E35.

42. Buuren Sv. Flexible Imputation of Missing Data. New York: Chapman and Hall/CRC; 2018.

43. Pritchard C, Wallace MS. Comparing UK and Other Western Countries' Health Expenditure, Relative Poverty and Child Mortality: Are British Children Doubly Disadvantaged? Child Soc. 2015;29(5):462-72.

---

## [Decision Letter · Decision Letter 1]

13 May 2020

PONE-D-19-19419R1

Recent quantitative research on determinants of health in high income countries: A scoping review

PLOS ONE

Dear Ms. Varbanova,

Thank you for submitting your manuscript to PLOS ONE. After careful consideration, we feel that it has merit but does not fully meet PLOS ONE’s publication criteria as it currently stands. Therefore, we invite you to submit a revised version of the manuscript that addresses the points raised during the review process.

Please address all comments raised by the reviewers.

 Specifically please not that the English in the present manuscript is not of publication quality and require major improvement. Please carefully proof-read and eliminate grammatical errors.

We would appreciate receiving your revised manuscript by Jun 27 2020 11:59PM. To enhance the reproducibility of your results, we recommend that if applicable you deposit your laboratory protocols in protocols.io, where a protocol can be assigned its own identifier (DOI) such that it can be cited independently in the future. For instructions see: http://journals.plos.org/plosone/s/submission-guidelines#loc-laboratory-protocols

We look forward to receiving your revised manuscript.

Kind regards,

Amir Radfar, MD,MPH,MSc,DHSc

Academic Editor

PLOS ONE

Reviewers' comments:

Reviewer's Responses to Questions

**Comments to the Author**

1. If the authors have adequately addressed your comments raised in a previous round of review and you feel that this manuscript is now acceptable for publication, you may indicate that here to bypass the “Comments to the Author” section, enter your conflict of interest statement in the “Confidential to Editor” section, and submit your "Accept" recommendation.

Reviewer #3: All comments have been addressed

Reviewer #4: (No Response)

2. Is the manuscript technically sound, and do the data support the conclusions?

Reviewer #3: Partly

Reviewer #4: Yes

3. Has the statistical analysis been performed appropriately and rigorously? 

Reviewer #3: Yes

Reviewer #4: Yes

4. Have the authors made all data underlying the findings in their manuscript fully available?

Reviewer #3: Yes

Reviewer #4: Yes

5. Is the manuscript presented in an intelligible fashion and written in standard English?

Reviewer #3: No

Reviewer #4: Yes

6. Review Comments to the Author

Reviewer #3: - English Language, syntax and punctuation needs substantial improvement. Don’t use long sentences.

- The 2nd paragraph of introduction has not been written in scientific language. Needs substantial change.

- How this review and focusing on methods, can benefit the readership? Appraisal of methods could be a part of big review.

- Page 5, line 73, please start a new paragraph.

- The added sentences from line 83-89 is suitable for discussion not introduction.

- Please remove lines 89-91.

- In page 7, please put search strategy in a table, not in the text.

- Who developed and performed the search strategy? Was a librarian consulted? What is PICO?

- Line 134, please provide Kappa statistics, and intra and inter rater agreements

- Line 137, how do you define “aggregate level factor”?

- Line 151, the single person used for date extraction does not seem reliable. It is suggested that at least 2 persons should independently extract data, or the single data extractor be calibrated previously.

- Page 8, please include and describe data analysis method utilized in your study

- line 552-554, please modify this statement. This is broad and you have only focused on methods.

- Considering the focus of the review on methods, the discussion needs to be richer and critically appraise the findings with regards to methods (not studies, as justified in line 596). How the findings of these review can be helpful? Etc…

- I believe some explanations provided for past reviews can be implemented into the text for the convenient of reads too. Please do so when possible.

Reviewer #4: First of all, thanks to the authors for the work presented and for the chosen theme. Then, some suggestions, if they consider their pertinence.

I suggest that the search string be added to the supplementary material of the manuscript.

The footnotes may be show at the appendix file where it is located the figure 2. At time, please, review figure 2 and add the title.

If possible, try to reduce the number of pages and focus on the primary objective of the manuscript.

7. PLOS authors have the option to publish the peer review history of their article (what does this mean?). If published, this will include your full peer review and any attached files.

Reviewer #3: No

Reviewer #4: No

---

## [Author Response · Author response to Decision Letter 1]

29 May 2020

Please see document "Response to Reviewers".

---

## [Decision Letter · Decision Letter 2]

30 Jul 2020

PONE-D-19-19419R2

Recent quantitative research on determinants of health in high income countries: A scoping review

PLOS ONE

Dear Dr. Varbanova,

Thank you for submitting your manuscript to PLOS ONE. After careful consideration, we feel that it has merit but does not fully meet PLOS ONE’s publication criteria as it currently stands. Therefore, we invite you to submit a revised version of the manuscript that addresses the points raised during the review process.

ACADEMIC EDITOR:  The manuscript has been improved significantly. Please address the comments made by reviewer # 3 

We look forward to receiving your revised manuscript.

Kind regards,

Amir Radfar, MD,MPH,MSc,DHSc

Academic Editor

PLOS ONE

Reviewers' comments:

Reviewer's Responses to Questions

**Comments to the Author**

1. If the authors have adequately addressed your comments raised in a previous round of review and you feel that this manuscript is now acceptable for publication, you may indicate that here to bypass the “Comments to the Author” section, enter your conflict of interest statement in the “Confidential to Editor” section, and submit your "Accept" recommendation.

Reviewer #3: All comments have been addressed

Reviewer #5: All comments have been addressed

2. Is the manuscript technically sound, and do the data support the conclusions?

Reviewer #3: Yes

Reviewer #5: Yes

3. Has the statistical analysis been performed appropriately and rigorously? 

Reviewer #3: Yes

Reviewer #5: N/A

4. Have the authors made all data underlying the findings in their manuscript fully available?

Reviewer #3: Yes

Reviewer #5: Yes

5. Is the manuscript presented in an intelligible fashion and written in standard English?

Reviewer #3: Yes

Reviewer #5: Yes

6. Review Comments to the Author

Reviewer #3: - In abstract background, provide aim of the study.

- In the abstract methods section, please give more details about methods and move the results part “Sixty studies that 16 performed cross-national statistical analyses aiming to evaluate the impact of one 17 or more aggregate level determinants on one or more general population health 18 outcomes in high-income countries were selected” to results.

- In abstract result, please modify last sentence as “effects fluctuated between statistically significant and not significant, and between beneficial to health and detrimental to health”

- Merge the answer to comment “How this review and focusing on methods, can benefit the readership? Appraisal of methods could be a part of big review.” To the first paragraph of discussion.

- Mention the answer to “Who developed and performed the search strategy? Was a librarian consulted? What is PICO?” in the methods section.

- The reviewer intention from comment “I believe some explanations provided for past reviews can be implemented into the text for the convenient of reads too. Please do so when possible” was to ask the authors to include their explanations for reviewers in the past round of review into the manuscript. The authors have provided some good explanations in the past round of review which can benefit readers too. So please add your explanations to reviewers in the past round of review to the text when possible.

Reviewer #5: The authors have addressed all concerns raised in the initial review. The manuscript has been improved substantially.

7. PLOS authors have the option to publish the peer review history of their article (what does this mean?). If published, this will include your full peer review and any attached files.

Reviewer #3: No

Reviewer #5: No

---

## [Author Response · Author response to Decision Letter 2]

14 Aug 2020

Please see file named "Response to Reviewers".

---

## [Decision Letter · Decision Letter 3]

31 Aug 2020

Recent quantitative research on determinants of health in high income countries: A scoping review

PONE-D-19-19419R3

Dear Dr. Varbanova,

We’re pleased to inform you that your manuscript has been judged scientifically suitable for publication and will be formally accepted for publication once it meets all outstanding technical requirements.

Kind regards,

Amir Radfar, MD,MPH,MSc,DHSc

Academic Editor

PLOS ONE

Additional Editor Comments (optional):

Reviewers' comments:

Reviewer's Responses to Questions

**Comments to the Author**

1. If the authors have adequately addressed your comments raised in a previous round of review and you feel that this manuscript is now acceptable for publication, you may indicate that here to bypass the “Comments to the Author” section, enter your conflict of interest statement in the “Confidential to Editor” section, and submit your "Accept" recommendation.

Reviewer #3: All comments have been addressed

Reviewer #6: All comments have been addressed

2. Is the manuscript technically sound, and do the data support the conclusions?

Reviewer #3: Yes

Reviewer #6: Yes

3. Has the statistical analysis been performed appropriately and rigorously? 

Reviewer #3: Yes

Reviewer #6: I Don't Know

4. Have the authors made all data underlying the findings in their manuscript fully available?

Reviewer #3: Yes

Reviewer #6: Yes

5. Is the manuscript presented in an intelligible fashion and written in standard English?

Reviewer #3: Yes

Reviewer #6: Yes

6. Review Comments to the Author

Reviewer #3: (No Response)

Reviewer #6: I reviewed the comments and answers . I believe all previous comments have been appropriately addressed.

7. PLOS authors have the option to publish the peer review history of their article (what does this mean?). If published, this will include your full peer review and any attached files.

Reviewer #3: No

Reviewer #6: No

---

## [Editor Report · Acceptance letter]

4 Sep 2020

PONE-D-19-19419R3 

Recent quantitative research on determinants of health in high income countries: A scoping review 

Dear Dr. Varbanova:

I'm pleased to inform you that your manuscript has been deemed suitable for publication in PLOS ONE. Congratulations! Your manuscript is now with our production department. 

Kind regards, 

on behalf of

Dr. Amir Radfar 

Academic Editor

PLOS ONE